# Costs of services and funding gap of the Bangladesh National Tuberculosis Control Programme 2016–2022: An ingredient based approach

Md. Zahid Hasan [1,2]*, Sayem Ahmed [3], Zeenat Islam[1], Farzana Dorin[1], Md. Golam Rabbani[1], Gazi Golam Mehdi[1], Mohammad Wahid Ahmed [1], Tazeen Tahsina[4], Shehrin Shaila Mahmood [1], Ziaul Islam[1]

1 Health Economics and Financing, Health Systems and Population Studies Division, International Centre for Diarrhoeal Disease Research, Bangladesh (icddr,b), Dhaka, Bangladesh, 2 Leeds Institute of Health Sciences, University of Leeds, Leeds, United Kingdom, 3 Health Economics and Health Technology Assessment (HEHTA), Institute of Health & Wellbeing, University of Glasgow, Glasgow, United Kingdom, 4 Maternal and Child Health Division, International Centre for Diarrhoeal Disease Research, Bangladesh (icddr,b), Dhaka, Bangladesh

* md.zahid@icddrb.org, m.zahidhasan3@gmail.com

**Data Availability Statement:** All relevant data are within the paper and its Supporting Information files.

## Abstract

### Background

Bangladesh National Tuberculosis (TB) Control Programme (NTP) has deployed improved diagnostic technologies which may drive up the programme costs. We aimed to estimate the supply-side costs associated with the delivery of the NTP and the funding gap between the cost of implementation and available funding for the Bangladesh NTP.

### Methods

An ingredient-based costing approach was applied using WHO's OneHealth Tool software. We considered 2016, as the base year and projected cost estimates up to 2022 using information on NTP planned activities. Data were collected through consultative meetings with experts and officials/managers, review of documents and databases, and visits to five purposively selected TB healthcare facilities. The estimated costs were compared with the funds allocated to the NTP between 2018 and 2022 to estimate the funding gap.

### Findings

The estimated total cost of NTP was US$ 49.22 million in 2016, which would increase to US$ 146.93 million in 2022. Human resources (41.1%) and medicines and investigations/ supplies (38.0%) were the major two cost components. Unit costs were highest for treating extensively drug-resistant TB at US$ 7,422.4 in 2016. Between 2018–2022, NTP would incur US$ 536.8 million, which is US$ 235.18 million higher than the current allocation for NTP.

**Funding:** This work was supported by the United States Agency for International Development (USAID) under the terms of the USAID's Research for Decision Makers (RDM) Activity cooperative agreement [AID-388-A-17-00006]. icddr,b acknowledges with gratitude the commitment of USAID's RDM activity to its research efforts and funding for this study. The funders had no role in study design, data collection and analysis, decision to publish, or preparation of the manuscript.

**Competing interests:** All other authors declare no conflicts of interest.

## Conclusion

Our results indicated a funding gap associated with the NTP in each of the years between 2018–2022. Policy planners should advocate for additional funding to ensure smooth delivery of TB services in the upcoming years. The cost estimates of TB services can also be used for planning and budgeting for delivering TB services in similar country contexts.

## Introduction

Tuberculosis (TB) is the 13[th] leading cause of death and second most infectious condition after COVID-19 [1]. Despite being treatable and curable, TB infected an estimated 10 million people and killed 1.5 million people in 2020; and is therefore, a threat to global public health [2]. Geographically, most people who developed TB in 2020 were in the WHO regions of Southeast Asia (43%) and three countries from here had 38% of the global TB cases. The incidence of TB is much higher in low-and-middle income countries (LMICs) like Bangladesh [3].

The estimated incidence of TB is 221 per 100,000 population in Bangladesh, with a mortality rate of 24 per 100,000 population [4]. Despite the availability of preventive and curative healthcare, the disease continues to be a threat to Bangladesh since the country accounts for 4% of the global TB burden and the seventh in identification of TB cases [3]. The country experiences nine deaths per hour due to TB with an annual death of 73,000 people [5]. Furthermore, every year around 10,000 cases of Multi-Drug Resistance TB (MDR TB) are identified [6]. Bangladesh National Tuberculosis Control Programme (NTP) is responsible for policy, planning, management, co-ordination, training, monitoring, and implementation of TB services. NTP collaborates with several national and international health and development organizations to implement the TB strategy in Bangladesh. A major part of TB funding for the NTP in Bangladesh comes from the Global Fund to Fight AIDS, Tuberculosis, and Malaria (GFATM). TB patients receive free treatment; and, in certain cases, such as patients with MDR TB, incentives are offered to encourage patients to continue treatment.

Examining the cost of providing healthcare services to TB patients is essential for effective planning and efficient use of available healthcare resources. Several studies have been conducted to estimate the cost of TB interventions in different countries [7]. For example, in India the provider cost (including costs for drugs, investigations, and shared costs) per patient was estimated between US$30 and US$43 [8], in Iran the average treatment cost of a patient with drug-sensitive TB ranged from US$34 –US$12,800 [9]. However, such evidence on provider costs of TB services is limited in Bangladesh.

In recent years, NTP has introduced different diagnostic technologies such as molecular testing (GeneXpert, Line probe assay) and revised diagnostic/screening algorithm aiming to increase the use of modern technologies to accelerate TB case detection. The organization has also revised treatment regimens for retreatment cases and introduced broader preventive therapy to effectively manage and prevent TB cases [10]. Such initiatives are likely to require the scaling-up of high-quality drug-sensitive, and drug-resistant (DR), TB treatment. The resources required for providing such treatments may go far beyond any previous efforts of NTP in allocating funds for TB services. Introduction, expansion, and maintenance of these new technologies have additional cost implications. Furthermore, in the context of the country's current LMIC status, it is anticipated that donor funding support will gradually decrease in the coming years and the issue of domestic financing will emerge. However, to date no study has been conducted in Bangladesh to estimate the supply side costs of TB services. Such

information is essential for evidence-based planning and decision making to ensure sustainability of TB services, especially as external funding reduces. To address the dearth of cost information of TB services e.g., unit cost of interventions, we aim to examine the supply side costs of TB services and funding gap between the costs of implementation and available funding for Bangladesh NTP using OneHealth Tool (OHT) [11].

## Materials and methods

### Study design and setting

This was a cross-sectional study applying an ingredient-based costing approach using WHO's customized OHT software. This tool has been widely used in planning and budgeting resources for health sector [12]. Bangladesh NTP provides services to TB patients through public (e.g., primary, secondary, and tertiary level facilities, and chest disease clinics and hospitals), private (e.g., hospitals and clinics), and non-government organization (NGO) facilities (e.g., BRAC, Damien, and others). We collected facility-related data from twelve purposively selected facilities representing different levels of diagnosis and treatment offered by the government and NGOs services providers. The facilities selected were from four main types of providers: 1) DOTs Centres (government and NGO), 2) chest disease clinics (CDC), 3) chest disease hospitals (CDH) (government and NGO), and 4) national and regional TB reference laboratories (S1 Table). These facilities were selected in consultation with the NTP as representative of facilities providing TB care in Bangladesh. Facility-level information and treatment procedures for TB patients were collected from each facility via examination of physical inventories and interviews with key personnel.

### Costing approach

In Bangladesh, public, private, and NGOs providers deliver TB services through different layers of the health system. We customized the OHT in the context of the Bangladeshi health system by forming a Technical Working Group (TWG) that includes representatives from NTP and personnel from public, private, and NGO providers of TB care. In consultation with the TWG, we selected available TB services (e.g., diagnostics or treatment) and assumed five service delivery channels for estimating the costs of TB services: i) Community (Preventive therapy and awareness campaign), ii) DOT Centers/corners (hospital outpatient DOTS and domiciliary DOTs), iii) Microscopy centers (sputum microscopy at health facility level), iv) Hospitals (providing TB inpatient, X-ray, Culture, DST & LPA), v) X-pert centers (GeneXpert test at facility level). We assumed that the selected TB interventions (Box 1) were being delivered through these five delivery channels. The list of interventions (investigations and treatments) considered for costing is detailed in Box 1. The interventions included several tests for TB diagnosis namely, Microscopy, X-ray, Culture, Drugs susceptibility testing (DST), GeneXpert, LPA (Line Probe Assay), FNAC (Fine Needle Aspiration Cytology), and Mendel–Mantoux test [13]. Additionally, few supplementary tests are done for the baseline and follow-up of MDR-TB treatment, such as Audiometry, ECG, Complete Blood Count, Serum Creatinine, Serum Electrolyte, Serum Bilirubin, Alkaline Phosphatase, Thyroid Function Test (TSH) [14]. The interventions related to treatment include 'preventive therapy' [15], treatment for 'drug susceptible TB' patients [13], and 'drug-resistant TB' patients [14].

However, it is worth noting that the service delivery channels outlined above are different from the facility structure of Bangladesh. For example, an Upazila Health Complex (UzHC) provides both DOTs and microscopy services. Consequently, we considered UzHC as two separate delivery channels, i.e., DOTs centre and Microscopy centre. We used three modules of

## Box 1. TB investigations and treatments considered for costing.

| **Investigations** |
| --- |

1. Microscopy
2. X-rays
3. Culture
4. Drugs susceptibility testing
5. GeneXpert
6. Resistance testing LPA (Line Probe Assay): For first-line drugs, previously treated TB cases **(From 2019)**
7. Resistance testing with LPA: For second-line drugs **(From 2019)**
8. FNAC (Fine Needle Aspiration Cytology)
9. Mendel–Mantoux test

**Other investigations** (for MDR TB baseline and follow-up)
10. HIV test, 11. Audiometry, 12. ECG, 13. Complete Blood Count, 14. Serum Creatinine, 15. Serum Electrolyte, 16. Serum Bilirubin, 17. Serum Glutamic-Pyruvic 18. Transaminase (SGPT), 19. Alkaline Phosphatase, 20. Blood Glucose, 21. Thyroid Function Test (TSH), and 22. Pregnancy Test.

| **Treatment** |
| --- |

**Regimen**
**Drug susceptible TB**
1. Initial treatment for Adult (Cat I)
2. Initial & previously treated treatment for children

**Started from 2019**
**Preventive therapy**
1. 3 HP for children 2 to 11 years old
2. 3 HP for adult and children more than 12 Years of age
3. IPT for children <2 years of age
**Drug susceptible TB**
1. New regimen for child retreatment cases (Pulmonary positive)
2. New regimen for child retreatment cases (Pulmonary negative)
3. New child retreatment cases- Meningitis, bone, Neurological TB
4. New regimen for adult retreatment cases (Pulmonary positive/Extrapulmonary TB)
5. New regimen for adult retreatment cases (Pulmonary negative)
6. New regimen for adult retreatment cases- Meningitis, bone, and Neurological TB
**Drug resistant TB**
1. MDR shorter regimen (9 months)
2. MDR longer regimen (20 months)
3. MDR with additional resistance (6 months Bedaquiline regimen)

**Discontinued from 2019**
**Preventive therapy**
1. IPT children under 5 years of age (up to 2019)
**Drug susceptible TB**
1. Previously treated for Adult (Cat II) (up to 2019)
**Drug resistant TB**
1. Shorter treatment regimen for MDR-TB
2. Longer treatment regimen for MDR-TB
3. XDR TB Regimen

the OHT to estimate the costs of selected TB interventions, i.e., health services module, health system module, and TB Impact Module and Estimates (TIME) (Fig 1).

After finalizing the interventions and delivery channels, we identified the ingredients used in delivering TB services, e.g., drugs, supplies, logistics, and duration of the treatment to estimate the unit costs of services at different delivery channels. The unit costs of services did not include the costs of human resources. We estimated the human resource cost separately using the health system module of OHT. The cost for ancillary drugs of TB services, specimen transportation, social support, and providers support were estimated separately.

The total cost of an intervention was estimated as follows:

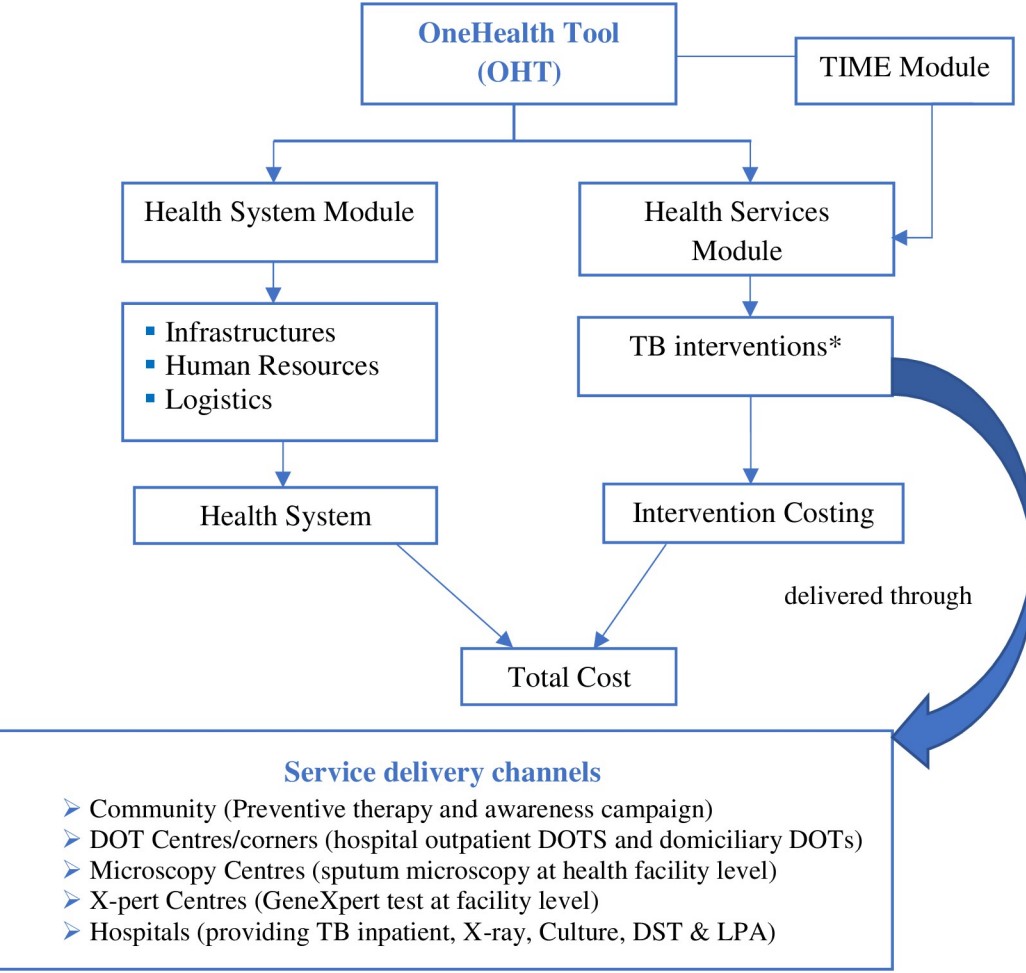

**Fig 1. Modules of the OHT and costing approach.** *List of TB interventions (Investigations and treatments) are given in Box 1.

*TB intervention cost = number of cases (of a given TB intervention) × unit of resources needed per case (excluding staff) × price per unit of resource*

The total cost of TB interventions for a delivery channel was estimated based on the proportion of interventions delivered through that delivery channel, determined as follows:

*Total cost per service delivery channel = number of cases (of a given TB intervention) × % availability of the service delivery channel*

The costs of infrastructure, human resources, and logistics were estimated using the health system module of OHT. In a similar way to the delivery channels used for costing service delivery, seven types of facilities providing TB diagnosis and healthcare services were identified to assist with capturing infrastructure costs. These were 1) DOTs Centres, 2) Microscopy Centre, 3) X-pert Centre, 4) X-ray Centre, 5) Chest disease clinic (CDC), 6) Chest disease hospital (CDH), and 7) Culture and Drug Susceptibility Testing (DST) Centre. The seven facility types were defined based on the services offered by existing healthcare facilities. Each facility

included in the study was categorised as one of these seven types. However, because some facilities offered services relating to more than one facility type, the related service components were separated for each type and costed separately. For example, at some CDCs both X-pert and Microscopy services are available, but in others only Microscopy services are available. So, we separately considered Microscopy Centre and X-pert Centre from CDCs, and excluded related resources (e.g., human resources, equipment, furniture, and supplies) while costing for CDCs to avoid double counting (Table 1).

We used information on healthcare facilities, i.e., size, equipment, supplies, and furniture for estimating costs of planned infrastructure. The costs of existing land used for buildings and existing infrastructure being used were not considered in the costing under the OHT. However, operation (e.g., maintenance, water, electricity) costs of these facilities were included. The cost of infrastructure was estimated only for the planned construction of facilities during the projected years 2017–2022. Cost data included all the capital and recurrent items from the program perspective. The cost of human resources was estimated using salary levels, increments, and benefits. In addition to TB associated human resources under government facilities, we included human resources from two large NGOs that provide TB services: BRAC and Damien. The logistics cost was estimated using information from NTP's central warehouse.

## TIME calibration

We projected the number of cases to be diagnosed, identified, and treated from the base year-2016 to the target years 2017–2022 based on the country's historical TB disease burden and treatment algorithms in the TIME module. We used historical data of TB incidence, current diagnostic algorithms and expected coverage of the diagnostic algorithms to produce estimates of the number of: TB tests to be performed, different types of TB cases to be identified e.g., new or retreatment cases, MDR cases, and MDR with additional resistance /pre-XDR cases. The estimates of different types of TB cases and tests derived from TIME are linked with the OHT module, allowing for more accurate estimates of TB programme costs. Online supplemental tables show the method for the TIME calibration in detail (S3–S6 Tables).

## Data collection

The key activities of the study were conducted from January to September 2018. From the selected facilities, information on available human resources, salary information, TB-related drugs, ancillary drugs, supplies, logistics, equipment, and diagnostic services were collected.

**Table 1. TB services offered at different level of facilities.**

| Type of healthcare facility (Assumed) | Services offered |
|---|---|
| DOTs Centres | • Diagnostic microscopy<br>• Provision of TB Treatment (Outpatient and domiciliary level) |
| Microscopy Centre | Diagnostic microscopy |
| X-pert Centre | GeneXpert tests |
| X-ray Centre | X-ray |
| Chest disease clinic (CDC) | Provision of TB Treatment (Outpatient and domiciliary level) |
| Chest disease hospital (CDH) | • Line Probe Assay (LPA)<br>• Providing TB Treatment (Inpatient DOTS) |
| Culture and Drug Susceptibility Testing (DST) Centre | • Culture<br>• Drug Susceptibility Testing (DST) |

We used national TB treatment guidelines and consulted with national TB experts for treatment regimens (drugs and doses). We also consulted microbiologists to identify and quantify the required reagents and supplies that were used to perform different TB-related diagnostics (e.g., culture and DST). We collected the unit price of anti-TB drugs, supplies, and diagnostic reagents from the Global Drug Facility (GDF) product/medicine catalog 2016 and 2019 [16, 17]. Government sources were used for the price of supplementary diagnostic tests (e.g., ECG). We used Bangladesh NTP's diagnostic algorithms and information on case notification and treatment procedures from the 2017 NTP Annual Report (showing estimates for 2016) [18] and the United Nations High Level Meeting (UNHLM) targets set for 2018–2022 from the NTP [19].

## Data analysis

We estimated the annual costs for the base year of 2016 and the expected cost of target years from 2017–22 using OHT. Based on the available country-specific parameters and the treatment algorithm, the TIME model was calibrated (S1 File). All inputs used for providing TB services during the base year and expected to be used in the target years were identified, quantified, and valued in Bangladesh Taka (BDT) using checklists. The price was later validated with the local TB programme experts of NTP. The number of services (e.g., diagnostics and treatment) for the base year 2016 and those to be provided in the projected years 2017–2022 were estimated using the coverage information of OHT linked with the estimates from the TIME module. The estimated costs for the base year 2016 were adjusted for inflation in 2019 considering the inflation rates obtained from World Economic Outlook Database 2018 [20]. The exchange rates of BDT to US\$ in 2016 and mid-2019 were taken from the Bangladesh Bank and for the years 2020–22; Annual depreciation of 1.5% of BDT was considered based on the trend of the exchange rates of the past years [21, 22]. The new Levofloxacin-based treatment regimen,3HP preventive therapy, and new MDR shorter regimen (9 months), MDR longer regimen (20 months), and MDR with additional resistance (6 months Bedaquiline regimen) were implemented from 2019 (Box 1). Thus, the treatment cost per patient was re-estimated separately from July 2019 and onward.

All required data used for costing were reviewed by the study team and by the international OHT experts, for gaps, errors, duplications, and inconsistencies. After the necessary data validation, analysis was performed to produce results on diagnostic costs, treatment costs, unit cost by category of TB patient, and total cost (health services plus health system cost) of TB services provided by the Bangladesh NTP. We estimated the funding gap between the estimated costs for delivery of the NTP and the known funding allocated for its delivery in each year. As the National Strategic Plan (NSP) budgets are provided in the financial year (June to July), we converted this into calendar years and then compared them with the estimated costs using OHT for a similar period. As the NSP contained information from 2018–2022, we used this period for comparison.

## Ethical approval

The study didn't require ethical approval from the ethics committee. icddr,b research administration approved this as an activity (ACT-00901) that doesn't collect information from human participants.

**Table 2. Unit cost of TB services (investigation, treatment & prevention in 2016).**

| Services | Unit cost* (Drugs, reagents & supplies in 2016) | Inflation adjusted unit cost for 2019 |
|---|---|---|
| **Investigations** | US$ | US$ |
| Diagnosis by microscopy (Z-N 18% & LED 82%) | 0.29 | 0.33 |
| Follow up microscopy (Z-N 18% & LED 82%) | 0.29 | 0.33 |
| Z-N microscopy (100%) | 0.33 | 0.37 |
| LED microscopy (100%) | 0.29 | 0.33 |
| Culture (solid 80%, liquid 20%) | 4.1 | 4.7 |
| Drugs susceptibility test (solid 80%, liquid 20%) | 8.1 | 17.7 |
| Gene-Xpert test (100%) | 11.8 | 13.3 |
| X-ray chest (100%) | 4.4 | 5.0 |
| Fine Needle Aspiration Cytology (FNAC) | 1.9 | 2.1 |
| **Treatment (Anti-TB drugs plus baseline and follow-up tests)** | | |
| Category I (cost per adult patient): First-line TB treatment: Initial treatment for adult | 39.6 | 44.7 |
| Treatment for children (cost per child) | 21.2 | 23.9 |
| Category II: First-line TB treatment: Previously treated adult | 102.9 | 116.2 |
| Long regimen treatment for MDR TB (20 months)** | 1,775.2 | 2,002.5 |
| Short regimen treatment for MDR-TB (9 months)** | 1,243.1 | 1,402.3 |
| XDR TB treatment regimen (24 months)** | 7,422.4 | 8,372.9 |
| **Prevention** | | |
| IPT for children under 5 years of age | 30.2 | 41.3 |
| **New treatment for children (started from 2019)** | | |
| 3 HP for children 2 to 11 years old | | 30.2 |
| 3 HP for adult and children more than 12 Years of age | | 50.2 |
| IPT for children <2 years of age | | 23.3 |
| New regimen for child retreatment cases (Pulmonary negative) | | 109.3 |
| New child retreatment cases- Meningitis, bone, Neurological TB | | 188.4 |
| Levofloxacin based regimen for child retreatment cases (Pulmonary positive) | | 119.8 |
| **New treatment for adults (started from 2019)** | | |
| New Levofloxacin based regimen for Adult retreatment cases (Pulmonary positive/Extrapulmonary TB) | | 102.7 |
| **New treatment for MDR TB (started from 2019)** | | |
| New regimen for adult retreatment cases (Pulmonary negative) | | 84.4 |
| New regimen for adult retreatment cases- Meningitis, bone, and Neurological TB | | 156.3 |
| New MDR shorter regimen (9 months) | | 1,054.9 |
| New MDR longer regimen (20 months) | | 3,102.5 |
| New MDR with additional resistance (6 months Bedaquiline regimen) | | 4,749.7 |

*Exchange rate of 2016:1 US$ = BDT 78.3, 2019: 1 US$ = 84 BDT

**excluding cost of ancillary drugs, specimen transportation, social support and providers incentive but inclusive baseline and follow-up tests

## Results and discussion

### Unit costs of TB interventions

The lowest cost TB intervention was smear microscopy, costing US$ 0.3, whereas the cost per Gene-Xpert test was the highest at US$ 12. Based on the proportion of usage (solid 80%, liquid 20%),–the cost per culture was US$ 4. With similar proportions applied for DST solid and liquid- the unit cost of DST was US$ 8 in 2016 (inflation-adjusted figures for 2019 are shown in **Table 2**).

In 2016, first-line TB treatment (drugs and diagnostics) for an adult patient under category-I costs around US$ 40 and previously treated adult cases (category-II) around US$ 103. In the same year, drugs, and diagnostics per longer regimen MDR-TB patient costs around US$ 1,775, combining the cost of DOTs at the hospital and domiciliary level. The MDR-TB shorter regimen costs around US$ 1,243 per patient. The cost per XDR-TB patient was US$ 7,422 in 2016 and this was the highest among all TB cases.

Considering the 'new treatment regimen' and the 'diagnostic algorithms', the total cost per MDR patient under the longer regimen was estimated at US$ 3,103and the shorter 9-month regimen was US$ 1,054. The cost of this new shorter treatment regimen is approximately US$ 347 less than the previous shorter treatment regimen for MDR-TB in 2016.

Including algorithm changes in 2019, the total cost of drugs and diagnostics per MDR with additional resistance (6 months Bedaquiline regimen) patients was US$ 4,750 which was lower by US$ 2,326.8 compared to the cost of XDR patients in 2016. According to new preventive therapy (3HP), the total cost per adult and child aged 12 years and older (for 3 months weekly of Isoniazid and Rifapentin therapy) was estimated to be US$ 50.

Table 3 shows the estimated number of TB services to be delivered according to the TIME module in different years. A decreasing trend of TB diagnosis by microscopy was observed as per NTP plan. Contrary to the microscopy, an increasing trend of TB diagnosis by Gene-Xpert and chest X-ray was observed over the period. This predicted change in use of TB diagnostic tools was linked with greater resources committed to the purchase of Gene-Xpert and digital X-ray machines in the current operational plan of Bangladesh NTP. Resistance testing with LPA for 1[st] and 2[nd] line drugs depicted an increasing trend from 2019 onwards as it was emphasized in the revised diagnostic algorithms since 2019. According to the new treatment regimen for 2019, there would be no category-II treatment from 2019 onwards, which was reflected in the numbers. Instead of cat-II treatment, the number of new Levofloxacin based regimens for retreatment cases since 2019 has been estimated for 2019–2022. The number of patients receiving a long regimen of MDR treatment would be reduced, while the number of patients with MDR under new shorter regimen (9 months) would increase; reflecting scale up of the new shorter regimen between 2019–2022. A similar trend of decreasing numbers of MDR longer regimen (20 months) was observed during 2019–2022.

### Total health system costs

The estimated total cost of the NTP was US$ 49.22 million for 2016 which is expected to increase to US$ 146.93 million in 2022 given the availability of resources committed, attainment of service coverage, and planned use of fixed assets (Table 4). From 2016 to 2022, the NTP is estimated to have cost US$ 648.0 million. The highest proportion of the total cost is accounted for human resources (41.1%) followed by medicines, investigations/supplies including wastage (37.6%), and planned infrastructure (12.1%). Of the seven types of facility, construction cost was the highest for TB hospitals (S9 Table).

**Table 3. Total number of TB services of base year (2016) and projected years (2017–22) as per TB impact module.**

| Types of TB services | 2016 | 2017 | 2018 | 2019 | 2020 | 2021 | 2022 |
|---|---|---|---|---|---|---|---|
| Diagnosis microscopy | 3,670,204 | 3,652,448 | 3,627,988 | 3,639,596 | 3,633,140 | 3,632,070 | 3,594,588 |
| Follow Up microscopy | 454,606 | 471,548 | 492,837 | 503,857 | 511,212 | 513,733 | 513,331 |
| Diagnosis culture | 1,220 | 1,352 | 1,506 | 2,153 | 2,827 | 4,649 | 5,417 |
| Drug's susceptibility testing | 1,220 | 1,352 | 1,506 | 2,153 | 2,827 | 4,649 | 5,417 |
| Diagnosis GeneXpert | 10,783 | 26,798 | 58,604 | 72,118 | 97,270 | 123,111 | 172,000 |
| Resistance testing LPA: For first-line drugs, previously treated TB cases | - | - | - | 19,694 | 20,775 | 23,046 | 24,223 |
| Resistance testing with LPA: For second-line drugs | - | - | - | 2,131 | 2,799 | 4,603 | 5,363 |
| Diagnosis X-rays | 2,696 | 28,761 | 68,143 | 90,126 | 126,762 | 157,755 | 213,859 |
| First-line TB treatment: Initial treatment for Adult (Cat I) | 195,178 | 210,775 | 229,533 | 244,591 | 259,141 | 272,571 | 285,925 |
| First-line TB treatment: Initial & Previously treated treatment for children | 23,461 | 24,611 | 26,182 | 27,201 | 28,029 | 28,585 | 28,939 |
| First-line TB treatment: Previously treated for Adult (Cat II) | 13,963 | 15,208 | 16,652 | - | - | - | - |
| Lfx based regimen for child retreatment cases (Pul+) | - | - | - | 684 | 704 | 759 | 772 |
| IPT children under 5 years of age | 47,734 | 51,397 | 55,843 | - | - | - | - |
| 3 HP for children 2 to 11 years old | - | - | - | 59,444 | 71,889 | 84,923 | 98,718 |
| 3 HP for adult and children more than 12 Years of age | - | - | - | 141,533 | 164,746 | 188,718 | 213,888 |
| New regimen for child retreatment. cases (Pul-) | - | - | - | 361 | 371 | 401 | 408 |
| New child retrt. cases- Meningitis, bone, Neurological TB | - | - | - | 114 | 117 | 126 | 129 |
| New Lfx based regimen for Adult rtrt. Cases (Pul+ve/EPTB) | - | - | - | 8,141 | 7,654 | 7,450 | 6,734 |
| New regimen for adult retrt. cases (Pul-) | - | - | - | 5,694 | 6,023 | 6,700 | 7,065 |
| New regimen for adult retrt. cases- Meningitis, bone, and Nurological TB | - | - | - | 178 | 188 | 209 | 221 |
| IPT for children <2 years of age | - | - | - | 28,307 | 34,946 | 41,937 | 49,359 |
| Longer regimen for MDR-TB | 915 | 955 | 905 | - | - | - | - |
| Shortened nine-month treatment regimen for MDR-TB | 201 | 384 | 586 | - | - | - | - |
| XDR TB Regimen | 12 | 14 | 15 | - | - | - | - |
| MDR new shorter regimen (9 months) | - | - | - | 1,066 | 1,699 | 3,282 | 4,398 |
| MDR new longer regimen (20 months) | - | - | - | 1,066 | 1,100 | 1,321 | 965 |
| MDR with additional resistance (6 months Bedaquiline regimen) | - | - | - | 2,131 | 2,799 | 4,603 | 5,363 |
| HIV testing and counseling for TB patients | 63,700 | 61,629 | 59,096 | 55,551 | 51,244 | 46,761 | 42,155 |
| Audiometry | 1,220 | 1,352 | 1,506 | 2,153 | 2,827 | 4,649 | 5,417 |
| ECG | 1,220 | 1,352 | 1,506 | 2,153 | 2,827 | 4,649 | 5,417 |
| Complete Blood Count (CBC) | 1,220 | 1,352 | 1,506 | 2,153 | 2,827 | 4,649 | 5,417 |
| Serum Creatinine | 1,220 | 1,352 | 1,506 | 2,153 | 2,827 | 4,649 | 5,417 |
| Serum Electrolyte | 1,220 | 1,352 | 1,506 | 2,153 | 2,827 | 4,649 | 5,417 |
| Serum Bilirubin | 552 | 602 | 663 | 1,107 | 1,597 | 2,045 | 2,310 |
| SGPT | 1,220 | 1,352 | 1,506 | 2,153 | 2,827 | 4,649 | 5,417 |
| Alkaline Phosphatase | 552 | 602 | 663 | 1,107 | 1,597 | 2,045 | 2,310 |
| Blood Glucose | 552 | 602 | 663 | 1,107 | 1,597 | 2,045 | 2,310 |
| Thyroid Function Test (TSH) | 1,220 | 1,352 | 1,506 | 2,153 | 2,827 | 4,649 | 5,417 |
| Chest X-ray for Follow-up Investigations of MDR-TB Patients | 1,220 | 1,352 | 1,506 | 2,153 | 2,827 | 4,649 | 5,417 |
| Pregnancy Test | 552 | 602 | 663 | 1,107 | 1,597 | 2,045 | 2,310 |
| Fine Needle Aspiration Cytology (FNAC) | 8,732 | 7,835 | 6,810 | 5,437 | 3,836 | 2,014 | - |
| *Mantoux Test (MT)* | 47,734 | 51,397 | 55,843 | 229,283 | 272,580 | 314,530 | 361,964 |

The programme cost included freight, transport/vehicle, social support, provider incentives, advocacy campaigns, monitoring-supervision, repair & maintenance and comprised around 9.1% of the total cost (**Table 5**). Among the five selected delivery channels, DOTs

**Table 4. Total cost of TB programme including health systems cost in million US$: 2016–2022. (inflation adjusted for 2017–22).**

| Cost components | 2016 | 2017 | 2018 | 2019 | 2020 | 2021 | 2022 | Total |
|---|---|---|---|---|---|---|---|---|
| **Estimated costs using OHT** | US$ (% share) | | | | | | | |
| Human Resources | 29.71 (60.4) | 32.65 (52.7) | 35.37 (55.6) | 38.21 (42.2) | 40.85 (40) | 43.74 (32.7) | 46.21 (31.4) | 266.75 (41.2) |
| Infrastructure (medical equipment, renovation, construction of new facilities) | 4.95 (10.1) | 11.48 (18.5) | 7.58 (11.9) | 7.1 (7.8) | 7.01 (6.9) | 19.83 (14.8) | 20.51 (14) | 78.46 (12.1) |
| Medicines, supplies, and Investigations (including wastage) | 10.57 (21.5) | 12.19 (19.7) | 14.18 (22.3) | 35.85 (39.6) | 43.91 (43) | 58.88 (44) | 68.47 (46.6) | 244.04 (37.7) |
| Programme costs | 3.99 (8.1) | 5.67 (9.1) | 6.44 (10.1) | 9.3 (10.3) | 10.28 (10.1) | 11.37 (8.5) | 11.81 (8) | 58.85 (9.1) |
| **Million US$** | 49.22 (100) | 61.99 (100) | 63.58 (100) | 90.46 (100) | 102.05 (100) | 133.81 (100) | 146.99 (100) | 648.1 (100) |
| *US$ to BDT conversion rate** | *78.3* | *80.5* | *83.5* | *84.0* | *85.3* | *86.5* | *87.8* | *-* |

\* For 2020–22, per year 1.5% BDT depreciation was considered

centres accounted for the highest proportion of total cost (60%) followed by community-level delivery channels (24%) (**Table 6**).

## Financial space and funding gap

The funding gap between our estimates using OHT and NSP budget in 2018 was US$ 8.85 million. This will increase to US$ 81.0 million in 2022, with a total funding gap of US$ 235.18 million over the period of 2016–2022 (Table 7). The annual funding gap has been notable since 2019 due to the inclusion of new treatment regimens, diagnostic algorithms, and preventive therapy (Fig 2).

   We found that the NTP would have a funding gap of US$ 235 million to deliver the planned interventions from 2018–2022. Compared to the base year 2016, the estimated cost is expected to increase in the target years 2017–2022. This can be attributed to the variation in the cost of drugs, reagents, supplies, salaries of personnel, and other associated factors with delivering TB interventions. Bangladesh NTP is dependent on funding support from development partners namely, GFATM, USAID, WHO, and UNICEF. More than 80% of the funding comes from reimbursable projects via government and project aid directly from donors [23]. Estimates of

**Table 5. Programme management costs for TB programme in million US$.**

| Programme cost | 2016 | 2017 | 2018 | 2019 | 2020 | 2021 | 2022 |
|---|---|---|---|---|---|---|---|
| | US$ (% share) | | | | | | |
| Monitoring and Evaluation | 0.23 (5.7) | 0.24 (4.2) | 0.24 (3.8) | 0.26 (2.8) | 0.27 (2.6) | 0.28 (2.5) | 0.29 (2.5) |
| Communication, Media and Outreach | 0.28 (7.1) | 0.29 (5.2) | 0.3 (4.7) | 0.32 (3.4) | 0.33 (3.2) | 0.35 (3) | 0.36 (3) |
| Advocacy | 0.004 (0.1) | 0.01 (0.2) | 0.01 (0.2) | 0.01 (0.2) | 0.01 (0.1) | 0.02 (0.1) | 0.02 (0.1) |
| General Programme Management (Water, electricity, petrol and Oil, printing, Binding, Stationaries, seals & stamps, cleaning and washing) | 1.43 (35.9) | 1.74 (30.7) | 1.92 (29.8) | 2.07 (22.3) | 2.19 (21.3) | 2.28 (20.1) | 1.88 (15.9) |
| Other (collaborative TB/HIV activities, TB research, community involvement, overtime, postage, uniform, casual labour, freight cost and transport, social support for TB patients and incentive for provider) | 2.04 (51.2) | 3.39 (59.7) | 3.97 (61.6) | 6.63 (71.4) | 7.47 (72.7) | 8.44 (74.3) | 9.26 (78.5) |
| **Total** | 3.99 (100) | 5.67 (100) | 6.44 (100) | 9.3 (100) | 10.27 (100) | 11.37 (100) | 11.81 (100) |

**Table 6. Cost of drugs, supplies and investigations of TB interventions by delivery channels (excluding wastage and ancillary drugs) in million US$: 2016–22.**

| Delivery channels | 2016 | 2017 | 2018 | 2019 | 2020 | 2021 | 2022 | Total |
|---|---|---|---|---|---|---|---|---|
| | Million US$ (% share) | | | | | | | |
| Community | 1.22 (12.0) | 1.37 (11.7) | 1.53 (11.2) | 9.7 (28.1) | 11.91 (28.2) | 14.35 (25.3) | 17.03 (25.8) | 57.1 (24.3) |
| DOTS Centre | 7.07 (69.6) | 7.98 (68.1) | 8.91 (65.3) | 19.86 (57.6) | 24.13 (57.2) | 34.11 (60.2) | 38.6 (58.5) | 140.66 (59.9) |
| Microscopy Centre | 1.235 (12.2) | 1.28 (11.0) | 1.32 (9.7) | 1.4 (4.1) | 1.46 (3.5) | 1.52 (2.7) | 1.57 (2.4) | 9.78 (4.2) |
| Hospitals | 0.47 (4.6) | 0.66 (5.6) | 0.92 (6.8) | 2.25 (6.5) | 2.96 (7.0) | 4.38 (7.7) | 5.38 (8.2) | 17.02 (7.2) |
| X-pert centers | 0.17 (1.6) | 0.43 (3.7) | 0.96 (7.1) | 1.25 (3.6) | 1.76 (4.2) | 2.33 (4.1) | 3.38 (5.1) | 10.28 (4.4) |
| **Total** | 10.16 (100.0) | 11.71 (100.0) | 13.64 (100.0) | 34.47 (100.0) | 42.22 (100.0) | 56.69 (100.0) | 65.95 (100.0) | 234.85 (100.0) |

costs and funding gaps will be useful for advocating for additional funding for TB service delivery. The funding gap identified may be attributed to planned changes in the diagnostic algorithms, implementation of new diagnostic technologies, and use of new treatment regimens for treating TB patients.

We found that the unit cost of a Gene-Xpert test was US$ 12, which is higher compared to any other non-Gene-Xpert test e.g., microscopy. The finding is similar to a study conducted in Uganda that found the total diagnostic costs were higher in Xpert-based clinics compared to in non-Xpert clinics [24]. On the other hand, the newly developed drug susceptibility test (DST) was found to be less expensive (US$8) compared to Gene-Xpert test. However, it required more time to detect TB and MDR-TB cases using DST [25].

The current study showed that costs per patient with the shorter treatment regimens for MDR (9 months) and MDR with additional resistance were substantially lower compared to the other existing treatment regimens for MDR. This finding is supported by previous studies from different countries, including LMICs [26, 27]. However, the drugs and supplies costs of MDR-TB treatment in such countries were higher compared to Bangladesh—US$ 2,500–US$ 3,500 per patient in Peru and Philippines and US$ 9,000–US$ 10,000 per patient in Estonia and Russia [26–28]. This might be lower in Bangladesh because we did not consider the cost of anciliary drugs, human resources, social support, and transportation costs while calculating the unit costs.

We found that the introduction of the new treatment regimen and preventive therapy would increase the cost dramatically from 2018 to 2022. Consequently, the funding gap would increase. In line with this trend of high costs, some new technologies are associated with high expenditures, leading to greater investment in the health care programme for accurate identification and treatment of diseases [29, 30]. It is evident that although the TB services are free in Bangladesh, access to quality services, and modern screening facilities are major challenges in adequate detection and treatment of the TB cases [31]. Funding gap may enhance such challenges, increase catastrophic expenditure [32], and decrease case notification as happened in

**Table 7. Funding gap between estimated costs using OHT and NSP budget 2018–22 (calendar year).**

| Years | Estimated using OHT 2018–22 | NSP budget 2018–22 | Funding gap 2018–22 |
|---|---|---|---|
| | Million US$ | Million US$ | Million US$ |
| 2018 | 63.6 | 54.75 | 8.85 |
| 2019 | 90.5 | 56.99 | 33.51 |
| 2020 | 102.1 | 60.85 | 41.25 |
| 2021 | 133.8 | 63.23 | 70.57 |
| 2022 | 146.9 | 65.9 | 81.00 |
| **Total** | **536.8** | **301.72** | **235.18** |

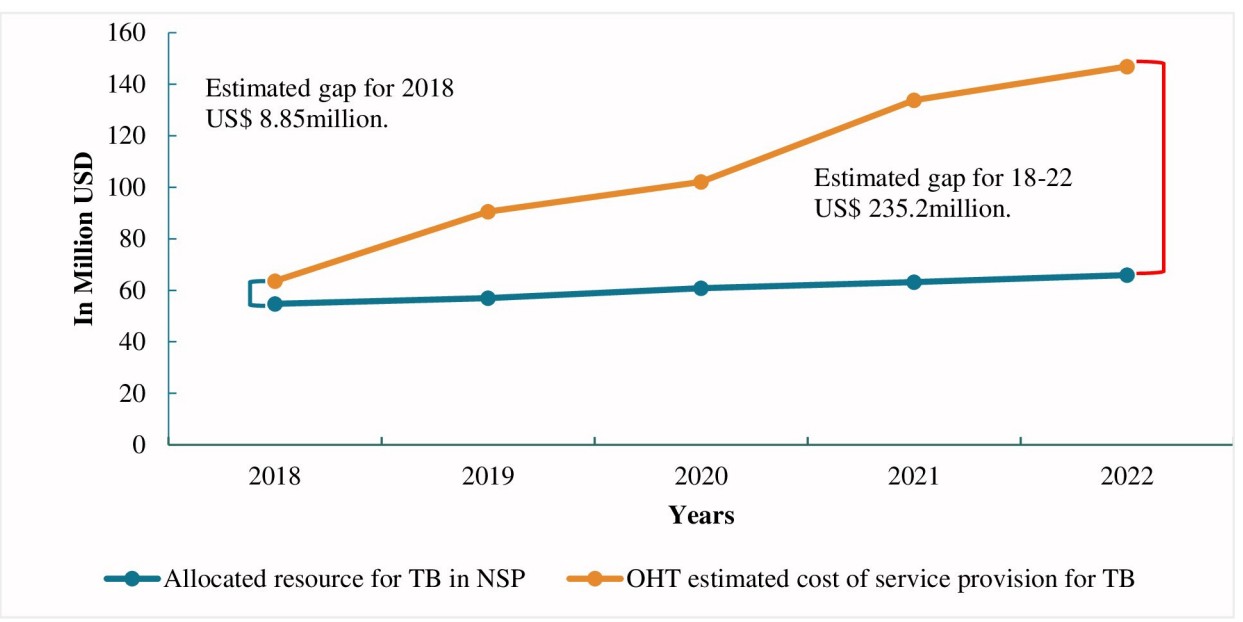

**Fig 2. Funding gap between OHT cost estimate and allocated budget in NSP 2018–22.**

2011 due to the funding gap from GFATM [33]. However, we did not study the consequences of underfunding during 2019–2022. It would be interesting to assess the consequences of the funding gap in future studies. The findings on the unit cost estimates from this study will be helpful for microlevel planning and budgeting of the Bangladesh NTP for efficient resource allocation and continued service delivery.

While the costing study provides valuable insights into the expenses associated with the TB services, it is important to note that the study has a few limitations that should be taken into consideration. First, the unit price of antituberculosis drugs (as shown in the Global Drug Facility's drug catalog) could vary depending on the volume of order, the requested delivery time, and the country's regulatory requirements. Second, based on the reported proportion (9–11%) of the total grant, the GFATM supported human resources costs at NTP headquarters were included in the total cost of human resources under the programme costs section. Third, assumptions were made for one of the warehouse system's related input price, quantities, and wastage as physical inventory was not possible. However, the assumptions were made based on the opinions of experts in this field and related to that warehouse. Fourth, we could not include HR costs of NGOs except for BRAC and Damien and private providers in our analysis. Besides government operated providers, BRAC and Damien are the two largest NGOs in providing TB services. However, this is the first study that included detailed information on the cost of different investigations and treatments, management of different channels of services, and estimated costs according to the recommended OHT of WHO guidelines in Bangladesh. The findings on the unit costs of intervention under this study could provide new insights and support evidence-based decisions in resource allocation in Bangladesh NTP and similar LMIC settings.

## Conclusion

The estimated funding gap between the OHT estimate and the 2018–22 NSP budget indicates the amount of additional investment required to deliver TB services effectively. Therefore,

policy planners should consider these estimates to advocate for increased funding to enable Bangladesh's NTP to match the costs. The cost estimates of TB services should be considered when preparing the next NSP for NTP. This study recommends scaling up the shorter regimen for MDR/MDR with additional resistance due to their lower costs compared to the earlier regimen. Based on the current estimates, further studies should delve into the cost of providing TB services with future changes in the diagnostic and treatment algorithms.

## Supporting information

**S1 Table. List of sites visited for data collection.**
(DOCX)

**S2 Table. Diagnostic algorithms.**
(DOCX)

**S3 Table. Sensitivity and specificity of different diagnostics.**
(DOCX)

**S4 Table. Coverage of diagnostic algorithms in care and control of OHT.**
(DOCX)

**S5 Table. Screening parameters.**
(DOCX)

**S6 Table. Description of different TB treatment regimen included in the costing.**
(DOCX)

**S7 Table. List of anti-TB drugs.**
(DOCX)

**S8 Table. List of List of supplies/ reagents.**
(DOCX)

**S9 Table. Cost of constructing new facilities by type of assumed units in 2016 US$.**
(DOCX)

**S1 File.**
(ZIP)

## Acknowledgments

The authors are indebted to Prof. (Dr.) Md. Shamiul Islam, Director of Mycobacterial Disease Control (MBDC) and Line Director TB-Leprosy & AIDS & STD programme (ASP), Ministry of Health and Family Welfare (MOHFW) for his guidance and support in conducting this study. The authors are thankful to Dr. Md. Assaduzzaman, Deputy Programme Manager, Bangladesh NTP, and Dr. Salim Hamid, National Advisor to NTP and Dr. Rupali Shishir Banu, National Consultant of NTP for their expert opinions and technical inputs. The authors also thankfully acknowledge the technical support provided by Dr. Md. Mojibur Rahman; former NTP consultant, Mustafizur Rahman, Microbiologist of Bangladesh NTP, Dr. Mursaleena Islam and Dr, Shamima Akhter of the Health Finance and Governance (HFG) project, and icddr,b's Dr. Sayera Banu (TB theme lead) and Dr. Shahriar Ahmed.

The authors would like to extend their special thanks to Ms. Nadia Carvalho, Carel Pretorious and Matt Hamilton for capacity building of the study team members and helping in customization of OneHealth Tool (OHT) and calibration of TB Impact & Estimate (TIME)

module. The authors are grateful to Bryony Dawkins, Lecturer in Health Economics, University of Leeds, for her input in improving clarity of the manuscript. Finally, the authors would like to express their gratitude to the participants of all key informant interviews, managers, and staff of the visited facilities for providing data to conduct this study. icddr,b is thankful to the Government of Bangladesh, Canada, Sweden and the UK for providing core/unrestricted support.

## Author Contributions

**Conceptualization:** Md. Zahid Hasan, Sayem Ahmed, Ziaul Islam.

**Data curation:** Md. Zahid Hasan, Sayem Ahmed, Mohammad Wahid Ahmed, Ziaul Islam.

**Formal analysis:** Md. Zahid Hasan, Sayem Ahmed, Zeenat Islam, Farzana Dorin, Md. Golam Rabbani, Gazi Golam Mehdi, Mohammad Wahid Ahmed, Ziaul Islam.

**Funding acquisition:** Ziaul Islam.

**Methodology:** Md. Zahid Hasan, Sayem Ahmed, Ziaul Islam.

**Project administration:** Ziaul Islam.

**Software:** Md. Zahid Hasan.

**Supervision:** Sayem Ahmed, Ziaul Islam.

**Validation:** Md. Zahid Hasan, Sayem Ahmed, Ziaul Islam.

**Visualization:** Md. Zahid Hasan, Ziaul Islam.

**Writing – original draft:** Md. Zahid Hasan, Sayem Ahmed, Zeenat Islam, Farzana Dorin, Md. Golam Rabbani, Gazi Golam Mehdi, Mohammad Wahid Ahmed, Tazeen Tahsina, Shehrin Shaila Mahmood, Ziaul Islam.

**Writing – review & editing:** Md. Zahid Hasan, Sayem Ahmed, Zeenat Islam, Farzana Dorin, Md. Golam Rabbani, Gazi Golam Mehdi, Mohammad Wahid Ahmed, Tazeen Tahsina, Shehrin Shaila Mahmood, Ziaul Islam.

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
