## [Decision Letter · Decision Letter 0]

7 Oct 2022

PONE-D-22-08101

Costs of services and funding gap of the Bangladesh National Tuberculosis Control Programme 2016-2022: An ingredient based approach

PLOS ONE

Dear Dr. Hasan,

Thank you for submitting your manuscript to PLOS ONE. After careful consideration, we feel that it has merit but does not fully meet PLOS ONE’s publication criteria as it currently stands. Therefore, we invite you to submit a revised version of the manuscript that addresses the points raised during the review process.

We look forward to receiving your revised manuscript.

Kind regards,

Kevin Schwartzman

Academic Editor

PLOS ONE

Journal Requirements:

Additional Editor Comments (if provided):

Thank you for submitting your work to PLOS ONE. The reviewers have raised a number of important points requiring revision. In particular, it is not clear why actual numbers of people diagnosed and starting treatment were not used for the years 2018-21, as opposed to the use of estimates from the TIME model to impute case numbers. Please address all reviewer comments carefully, and revise your manuscript accordingly. Thank you again.

Reviewers' comments:

Reviewer's Responses to Questions

**Comments to the Author**

1. Is the manuscript technically sound, and do the data support the conclusions?

Reviewer #1: Partly

Reviewer #2: Partly

2. Has the statistical analysis been performed appropriately and rigorously? 

Reviewer #1: Yes

Reviewer #2: Yes

3. Have the authors made all data underlying the findings in their manuscript fully available?

Reviewer #1: Yes

Reviewer #2: No

4. Is the manuscript presented in an intelligible fashion and written in standard English?

Reviewer #1: Yes

Reviewer #2: Yes

5. Review Comments to the Author

Reviewer #1: Thank you for the opportunity to read this manuscript. The study from Hasan et al, estimates the health system costs related to TB care in Bangladesh and its funding gap. It is a relevant manuscript, and a lot of work went into this, but I feel it is rather underreported. I have indicated a couple of places where more detail would be beneficial.

Introduction

1-Although PLOS ONE does not have a word count limit; it does recommend that authors report their findings concisely. That said, the introduction is considerable long, two full pages, and it is hard to follow. I suggest the authors short it up to four of five paragraphs.

2-Line 66- “Evidence on cost is not widely available on TB treatment”. This sentence is a bit vague, and it is not accurate. There are several publications on the costs of TB treatment, either on active TB or latent TB. I suggest rephrasing or removing the sentence.

3-Line 77- What were interventions put in place for DS and MDR-TB?

Methods

4- Selection of centres: Could the authors be more specific about why the five units were selected? The NTP suggested it, but what were the inclusion criteria used to choose these facilities. I would also recommend a table displaying the main characteristics of each health care unit. E.g. services offered: Micro lab, Xpert, Chest x-ray on-site, TB clinic or Family Health Clinic, Pharmacy etc. Furthermore, it would be interesting to see the number of active TB patients (DS-TB and MDR-TB) and the number of patients receiving TPT.

5-Maybe because I am not familiarized with the One Health Tool, I am having trouble following the description of the costing approach. I suggest that the authors create a diagram (or one figure) explaining the different modules of the OHT and which intervention falls in each of these modules.

6- Ingredients: I suggest the authors create a table listing the main ingredients that were valuated (which should go to the appendix). Currently, the authors cited in lines 123-127 big categories: TB drugs, logistics, supplies, social support etc. However, it is unclear which TB drugs, auxiliary drugs (e.g., pyridoxine, antiemetics?), and what type of supplies (culture media, GeneXpert cartridge).

7-TIME calibration: I did not follow why the authors had to estimate the number of TB patients for 2018-2022? Why did they not use the actual numbers up to 2021? Did the authors evaluate the robustness of the estimations? For instance, did they check if the estimations were similar to the number of TB cases reported by the NTP?

8- I suggest presenting the costs using only one n only one currency – preferably USD. It is not clear; now, the text has USD and BDT, some tables have USD and BDT, and others only BDT.

Results:

Minor comments: please define the abbreviations for all tables: for instance, what is FNAC in table 1?

9-Lines 209-214 and Table 1: What is the difference between the “Short regimen treatment for MDR-TB 9 months” (cost 1,403 in 2019) to the “MDR new short regimen” that cost 1,054.9.

I suggest a description of different regimes in a supplementary table or at least in the footnote of the tables. For instance, many readers may not be aware of the difference between cat I ad cat II regimes.

10-Table 3 and 5- please add the % per year not only for the total years.

Discussion:

Line 295- “This is the first study that included the information on the costs…”. I would delete this sentence. The authors did not provide any reference, and it is unlike that this was the first study evaluating the costs of investigations and treatment of TB – across the globe.

Line 299- I would be very cautious in saying that this study can be generalized to other LMICs, mainly because the healthy systems vary from country to country. Please rephrase or remove the sentence.

Reviewer #2: Hasan and colleagues estimated the cost of TB services and the existing funding gap in Bangladesh. This is a good use of the OHT and could be useful to the NTP, however the current manuscript does not make clear how such data will be used. I have some comments below.

Major Comment:

• My major comment has to do with the utility of this analysis—projecting costs for 2018-2022, but 2022 is halfway completed. How is the NTP funded? Are they given a fixed funding envelope and must work with it? Or do they fund services as they are needed? I think more could be said on how the NTP plans to use these data moving forward. The discussion repeated many results, but didn’t get into implications. This is needed. If the action item is that the NTP has been woefully underfunded for the last 5 years—and people who want care must pay out of pocket or will not receive care—then this must be said. If the action item is that the NTP has not been scaling infrastructure and performing as many tests, etc. as they said they would do—then this must be said. In sum, it is unclear what this funding gap represents. The authors should include a paragraph or two on how to interpret these findings and their implications.

Other comments:

• What is the case detection rate in Bangladesh? The assumption in OHT appears to be 100%, but WHO estimates 64%. Could this be where some of the “funding gap” is coming from? The fact that you are budgeting for 100% case detection, but only 64% are detected?

• “Geographically, most people who developed TB in 2020 were in the WHO regions of South-East Asia (43%) and eight countries from this region contain two thirds of the global TB cases.” – This statement appears to be contradictory. The region cannot account for only 43% of cases, but have eight countries accounting for 66%.

• How representative were the 5 selected facilities for surveying? When considering the NTP budget, do they also consider funds required for persons accessing care through private services or those through an NGO? The methods are unclear, and the selected facilities all appear public.

• The word logistics – I believe – is more commonly referred to as “overhead” in the literature.

• Was there consideration of annuitization for capital purchases – in particular for infrastructure?

• Can the authors provide insight into the “levofloxacin-based regimen”? This will not be totally evident to readers – ensure you specify what type of TB it is treating.

• Can the authors clarify in the results whether unit costs are inclusive of human resource needs (e.g., for treatment regimens)?

• What is the 6 months bedaquiline regimen for MDR? Is this BPaL? It is curious to see a regimen for additional resistance be the most frequently used regimen for MDR-TB.

• A list of infrastructure costs would be helpful to include. All underlying data should be made available.

6. PLOS authors have the option to publish the peer review history of their article (what does this mean?). If published, this will include your full peer review and any attached files.

Reviewer #1: No

Reviewer #2: No

---

## [Author Response · Author response to Decision Letter 0]

20 Feb 2023

Responses to reviewer’s comments

Reviewer 1

Thank you for the opportunity to read this manuscript. The study from Hasan et al, estimates the health system costs related to TB care in Bangladesh and its funding gap. It is a relevant manuscript, and a lot of work went into this, but I feel it is rather underreported. I have indicated a couple of places where more detail would be beneficial.

Introduction

Comment 1: 1-Although PLOS ONE does not have a word count limit; it does recommend that authors report their findings concisely. That said, the introduction is considerable long, two full pages, and it is hard to follow. I suggest the authors short it up to four of five paragraphs.

Response: Thank you for your comment. We have now revised the introduction as per your suggestion to make it clearer. (Please see page 1 and 2) 

Comment 2: 2-Line 66- "Evidence on cost is not widely available on TB treatment". This sentence is a bit vague, and it is not accurate. There are several publications on the costs of TB treatment, either on active TB or latent TB. I suggest rephrasing or removing the sentence.

Response: We have now revised the sentence according to your suggestion. “Several studies were conducted to estimate the cost of TB interventions in different countries. For example, in India the provider cost (including costs for drugs, investigations, and shared costs) per patient was estimated between US$30 and US$43 [13], in Iran the average treatment cost of a patient with drug-sensitive TB ranged from US$34 – US$12,800 [14].” (Please see page 3)

Comment 3: 3-Line 77- What were interventions put in place for DS and MDR-TB?

Response: We've included the new diagnostics tests put in place for DS and MDR-TB and changed it as below: 

"In recent years, NTP has introduced different diagnostic technologies such as molecular testing (GeneXpert, Line probe assay) which are likely require the scaling-up of high-quality drug-sensitive TB and drug-resistant (DR) TB treatment." (page 4, line 92-93)

 

Methods

Comment 4: 4- Selection of centres: Could the authors be more specific about why the five units were selected? The NTP suggested it, but what were the inclusion criteria used to choose these facilities. I would also recommend a table displaying the main characteristics of each health care unit. E.g. services offered: Micro lab, Xpert, Chest x-ray on-site, TB clinic or Family Health Clinic, Pharmacy etc. Furthermore, it would be interesting to see the number of active TB patients (DS-TB and MDR-TB) and the number of patients receiving TPT.

Response: Many thanks for your query on the selection of facilities. We wanted to select a representative of each type of facility for listing inventory and getting information on treatment procedure. As the structure of each type of facility are very similar, we selected five facilities in consultation with NTP to save time and resources for data collection. We have now included the types of services offered in different level of facilities. (Please see Table 1)

Comment 5: 5-Maybe because I am not familiarized with the One Health Tool, I am having trouble following the description of the costing approach. I suggest that the authors create a diagram (or one figure) explaining the different modules of the OHT and which intervention falls in each of these modules.

Response: We have now added a figure containing the different modul of the OHT and costing approach. Please see Figure 1 in the manuscript. 

Comment 6: 6- Ingredients: I suggest the authors create a table listing the main ingredients that were valuated (which should go to the appendix). Currently, the authors cited in lines 123-127 big categories: TB drugs, logistics, supplies, social support etc. However, it is unclear which TB drugs, auxiliary drugs (e.g., pyridoxine, antiemetics?), and what type of supplies (culture media, GeneXpert cartridge).

Response: We have now included two tables – 'List of anti-TB drugs' and 'List of supplies/ reagents' in the supplementary material. Please see S7 and S8 Table.

Comment 7: 7-TIME calibration: I did not follow why the authors had to estimate the number of TB patients for 2018-2022? Why did they not use the actual numbers up to 2021? Did the authors evaluate the robustness of the estimations? For instance, did they check if the estimations were similar to the number of TB cases reported by the NTP?

Response: Many thanks for this query. One of the main reasons for modelling the estimates on the number of services rather using information from 2018-2022 is that the information was not available for all the types of services used in costing during our data collection from January to September 2018. NTP regularly publishes annual report which mention information for previous year, for example the most recent TB annual report of 2020 includes information from 2019 and this is the latest available annual report. 

According to Annual NTP Report 2020, a total of 292,940 all forms of TB cases were reported in 2019. Our model estimate showed that 291,227 all forms of TB cases were under different types of treatment.

Comment 8: 8- I suggest presenting the costs using only one n only one currency – preferably USD. It is not clear; now, the text has USD and BDT, some tables have USD and BDT, and others only BDT.

Response: Many thanks for this suggestion. We have now presented the all tables and text using US$ value except Table 1 that contain some small amount in BDT and convertion may give wrong impression. 

Results:

Comment 9: Minor comments: please define the abbreviations for all tables: for instance, what is FNAC in table 1?

Response: We revised abbreviations in table and made the changes as below. 

Table 2: FNAC (Fine Needle Aspiration Cytology) 

Table 3: MT (Mantoux Test)

Comment 10: 9-Lines 209-214 and Table 1: What is the difference between the "Short regimen treatment for MDR-TB 9 months" (cost 1,403 in 2019) to the "MDR new short regimen" that cost 1,054.9.

I suggest a description of different regimes in a supplementary table or at least in the footnote of the tables. For instance, many readers may not be aware of the difference between cat I ad cat II regimes.

Response: We added a supplementary table including the description of different regimen as per your suggestion. (Please see Supplementary Table 6)

Comment 11: 10-Table 3 and 5- please add the % per year not only for the total years.

Response: We have now revised this accordingly. Please see Table 4 and Table 6

Discussion:

Comment 12: Line 295- "This is the first study that included the information on the costs…". I would delete this sentence. The authors did not provide any reference, and it is unlike that this was the first study evaluating the costs of investigations and treatment of TB – across the globe.

Response: Many thanks for pointing this. In Bangladesh, no comprehensive study has been done yet on the costing of different category TB treatment. We rephrased the sentence accordingly for the context of Bangladesh. 

"This is the first study that included detailed information on the cost of different investigations and treatments, management of different channels of services, and estimated costs according to the recommended OHT of WHO guidelines in Bangladesh." (page 15, line 348-350)

Comment 13: Line 299- I would be very cautious in saying that this study can be generalized to other LMICs, mainly because the healthy systems vary from country to country. Please rephrase or remove the sentence.

Response: Many thanks and we agree with your comment that despite similarity health system in LMIC vary from each other. We omitted 'generalization' from our statement and rephrased it as follows " The findings on the unit costs of intervention under this study could provide new insights and support evidence-based decisions in resource allocation in Bangladesh NTP and similar LMIC settings” (page 15, line 351-352)

 

Reviewer 2

Reviewer 2: Hasan and colleagues estimated the cost of TB services and the existing funding gap in Bangladesh. This is a good use of the OHT and could be useful to the NTP, however the current manuscript does not make clear how such data will be used. I have some comments below.

Major Comment: My major comment has to do with the utility of this analysis—projecting costs for 2018-2022, but 2022 is halfway completed. How is the NTP funded? Are they given a fixed funding envelope and must work with it? Or do they fund services as they are needed? I think more could be said on how the NTP plans to use these data moving forward. The discussion repeated many results, but didn't get into implications. This is needed. If the action item is that the NTP has been woefully underfunded for the last 5 years—and people who want care must pay out of pocket or will not receive care—then this must be said. If the action item is that the NTP has not been scaling infrastructure and performing as many tests, etc. as they said they would do—then this must be said. In sum, it is unclear what this funding gap represents. The authors should include a paragraph or two on how to interpret these findings and their implications.

Response: Many thanks for this query. NTP receives mixed type of funding i.e., from both government and donor organization. However, a major part of the funding comes from donor like global fund. Accodrding to operational plan of 2017-2022 about 83% of the funding to implement TB operational plan was as some type of development assistance. We have now added a few lines on NTP as follows 

“Bangladesh National Tuberculosis Control Programme (NTP) is responsible for policy, planning, management, co-ordination, training, monitoring, and implementation of TB services. NTP collaborates with several national and international health and development organizations to implement TB related strategy. A major part of TB funding for NTPs and other organizations in Bangladesh comes from the Global Fund to Fight AIDS, Tuberculosis, and Malaria (GFATM).” (Page 2, line 65-72)

We have also revised the discussion and added implication of fuding gap estimate as follows “Bangladesh NTP is dependent on the funding support from development partners namely, GFATM, USAID, and UNICEF. More than 80% of the funding comes from reimbursable project via government and project aid directly from donors [17]. Estimate on costs and funding gaps will be useful for advocating for additional funding to delivery planned TB services. This funding gap may be attributed to planned changes in the diagnostic algorithm, implementation of new diagnostic technologies, and use of new treatment regimens for treating TB patients.” (Page 13, line 310-316)

Other comments:

Comment: What is the case detection rate in Bangladesh? The assumption in OHT appears to be 100%, but WHO estimates 64%. Could this be where some of the "funding gap" is coming from? The fact that you are budgeting for 100% case detection, but only 64% are detected?

Response: Many thanks for this query. According to Annual Report 2020, the case detection rate in Bangladesh is 81% of total incidence of all forms of TB cases. The assumption of coverage algorithm is 100% which means that the target population was tested following any of the diagnostic algorithms and in total the coverage should sum to 100% as number of all tests should be included in the costing. As per plan of NTP, the coverage under x-pert was increasing which implied that more people would be diagnosed using x-pert in future. Dignosis of suspected cases may not imply that the cases are detected. However, we wanted to capture the costs of all diagnosis which were performed to detect TB cases following the diagnosis algorithm as shown in supplementary Table 2.

Comment: "Geographically, most people who developed TB in 2020 were in the WHO regions of South-East Asia (43%) and eight countries from this region contain two thirds of the global TB cases." – This statement appears to be contradictory. The region cannot account for only 43% of cases, but have eight countries accounting for 66%.

Response: We apologize for our contradictory statement. We revised and corrected the statement. "Geographically, most people who developed TB in 2020 were in the WHO regions of South-East Asia (43%) and three countries from this region contain 38% of the total global TB cases." (page 5, line 24)

Comment: How representative were the 5 selected facilities for surveying? When considering the NTP budget, do they also consider funds required for persons accessing care through private services or those through an NGO? The methods are unclear, and the selected facilities all appear public.

Response: Many thanks for raising this query. We apologies that there were some error in reporting the number of facilities, we have now updated this section. We have now added a para explaining how the facilities were considered for costing of insfrasturcture. To avoid complexity in analysis using OHT, we have assumed seven types of facilities that are providing TB diagnosis and healthcare services to the TB patients. This assumption was made based on the offered services of existing healthcare facilities. While doing so some facilities were actually split into more than one based on the offered services to avoide different group of same type of facility. For example, at some chest disease clinic (CDC), both Xpert and microscopy services are available and in some CDCs only microscopy services are available. So we separately considered microscopy centre and x-pert from CDCs and also excluded related resources to avoide double counting.

TB treatment in Bangladesh is provided at free of costs. Government provide the drugs and supplies to NGOs for treating patients. Some private providers offer TB services which are not considered in this costing. However, if a patient is diagnosed with TB at private provider they are most likely to refered to government providers for treatment, as the treatment here are free of cost. We have now included it as a limitation of this study.

Comment: The word logistics – I believe – is more commonly referred to as "overhead" in the literature.

Response: Many thanks for this comment. We have used ‘logistic’ as costs for ‘warehouses’ as defined in OHT.

Comment: Was there consideration of annuitization for capital purchases – in particular for infrastructure?

Response: This costing did not consider annuitization for purchase of capital items for infrastructure and we believe this is useful for estimating yearly equivalent costs for providing services. It is noted that the cost for existing infrastructure is not considered in the costing rather only operation costs of existing facilities is included. It is mentioned in the method section. Only the costs for infrasucture to be built over the projected period 2017-2022 was considered to project the required costs from programme perspective.

Comment: Can the authors provide insight into the "levofloxacin-based regimen"? This will not be totally evident to readers – ensure you specify what type of TB it is treating.

Response: We have added a supplementary table including the description of different regimen to clarify the treatment regimens. (Please see Supplementary Table 5)

Comment: Can the authors clarify in the results whether unit costs are inclusive of human resource needs (e.g., for treatment regimens)?

Response: We estimated the human resources cost using ‘health system module’ of OHT, separately. The unit costs of services did not include the costs for human resource. We mentioned this in the ‘Costing approach’ of 'Methods' section (Page 5, Line 122-124).

Comment: What is the 6 months bedaquiline regimen for MDR? Is this BPaL? It is curious to see a regimen for additional resistance be the most frequently used regimen for MDR-TB.

Response: We have added a supplementary table including the description of different regimen to clarify the treatment regimens. (Please see Supplementary Table 5)

Comment: A list of infrastructure costs would be helpful to include. All underlying data should be made available.

Response: We have now included the infrastructure costs as supplementary table S9.

---

## [Decision Letter · Decision Letter 1]

13 Mar 2023

PONE-D-22-08101R1Costs of services and funding gap of the Bangladesh National Tuberculosis Control Programme 2016-2022: An ingredient based approachPLOS ONE

Dear Dr. Hasan,

Thank you for submitting your manuscript to PLOS ONE. After careful consideration, we feel that it has merit but does not fully meet PLOS ONE’s publication criteria as it currently stands. Therefore, we invite you to submit a revised version of the manuscript that addresses the points raised during the review process.

Thank you for submitting your revised manuscript, which has addressed many of the earlier comments and concerns. However, reviewer 2 has highlighted several elements that still require further clarification. In addition, they have suggested careful proofreading and copyediting of the written English.

We look forward to receiving your revised manuscript.

Kind regards,

Kevin Schwartzman

Academic Editor

PLOS ONE

Journal Requirements:

Reviewers' comments:

Reviewer's Responses to Questions

**Comments to the Author**

1. If the authors have adequately addressed your comments raised in a previous round of review and you feel that this manuscript is now acceptable for publication, you may indicate that here to bypass the “Comments to the Author” section, enter your conflict of interest statement in the “Confidential to Editor” section, and submit your "Accept" recommendation.

Reviewer #1: All comments have been addressed

Reviewer #2: (No Response)

2. Is the manuscript technically sound, and do the data support the conclusions?

Reviewer #1: Yes

Reviewer #2: Partly

3. Has the statistical analysis been performed appropriately and rigorously? 

Reviewer #1: Yes

Reviewer #2: I Don't Know

4. Have the authors made all data underlying the findings in their manuscript fully available?

Reviewer #1: Yes

Reviewer #2: Yes

5. Is the manuscript presented in an intelligible fashion and written in standard English?

Reviewer #1: Yes

Reviewer #2: No

6. Review Comments to the Author

Reviewer #1: (No Response)

Reviewer #2: The authors have addressed many comments, but still some things remain. The manuscript would also benefit from English copyediting.

With respect to responses to reviewer comments:

Reviewer 1, Comment 3: I feel this was not properly explained in response and could be made clearer in the text with diagnostics and treatments referenced. Box 1 would benefit from some separation for tests/treatments specific to TB infection, drug-susceptible TB disease, and drug-resistant TB disease.

Reviewer 1, Comment 4; Reviewer 2: The authors did not address this comment adequately. The question had to do with selection of centers and the description is more confusing that before. I am not entirely clear on the procedures.

Reviewer 1, Comment 8: I think sticking to USD in all tables is advised.

Reviewer 2, Comment 1: It is still unclear what happened in the years where a funding gap existed. I think this comes back to using modeled estimates of resource needs and looking at what actually was done. The discussion would benefit from this information: what were the consequences of underfunding?

Other comments:

Line 67-72 -- perhaps simply refer to Global Health Cost Consortium (https://ghcosting.org/), which summarizes the available cost data for tuberculosis. You can also look to Value TB studies. The current summation is inadequate.

Lines 93-94 - unclear language "We collected facility-related data from purposively selected four types of twelve facilities representing different level of diagnosis and treatment of government and NGOs"

Results and Discussion -- these should be separate.

7. PLOS authors have the option to publish the peer review history of their article (what does this mean?). If published, this will include your full peer review and any attached files.

Reviewer #1: No

Reviewer #2: No

---

## [Author Response · Author response to Decision Letter 1]

3 May 2023

Reviewer 1, Comment 3: I feel this was not properly explained in response and could be made clearer in the text with diagnostics and treatments referenced. Box 1 would benefit from some separation for tests/treatments specific to TB infection, drug-susceptible TB disease, and drug-resistant TB disease.

Response: Many thanks for the comment. We have added the following lines as follows “The list of interventions (investigations and treatments) considered for costing is detailed in Box 1. The interventions included several tests for TB diagnosis namely, Microscopy, X-ray, Culture, Drugs susceptibility testing (DST), GeneXpert, LPA (Line Probe Assay), FNAC (Fine Needle Aspiration Cytology), and Mendel–Mantoux test [12]. Additionally, few supplementary tests are done for the baseline and follow-up of MDR-TB treatment, such as Audiometry, ECG, Complete Blood Count, Serum Creatinine, Serum Electrolyte, Serum Bilirubin, Alkaline Phosphatase, Thyroid Function Test (TSH) [13]. The interventions related to treatment include ‘preventive therapy’ [14], treatment for ‘drug susceptible TB’ patients [12], and ‘drug-resistant TB’ patients [13]. ” We have now updated the box as per suggestion. Please see page 5, line 121-128

Reviewer 1, Comment 4; Reviewer 2: The authors did not address this comment adequately. The question had to do with selection of centers and the description is more confusing that before. I am not entirely clear on the procedures.

Response: We apology for the confusion. We have now revised the section as follows “We collected facility-related data from twelve purposively selected facilities representing different levels of diagnosis and treatment offered by the government and NGOs service providers. The facilities selected were from four main types of providers: 1) DOTs Centres (government and NGO), 2) chest disease clinics (CDC), 3) chest disease hospitals (CDH) (government and NGO), and 4) national and regional TB reference laboratories (Supplementary Table 1). These facilities were selected in consultation with the NTP as representative of facilities providing TB care in Bangladesh.” Please see page 4, line 99-105

Reviewer 1, Comment 8: I think sticking to USD in all tables is advised.

Response: We have now USD in all tables. 

Reviewer 2, Comment 1: It is still unclear what happened in the years where a funding gap existed. I think this comes back to using modeled estimates of resource needs and looking at what actually was done. The discussion would benefit from this information: what were the consequences of underfunding?

Response: Many thanks for this query. It is evident that although the tuberculosis services are free, limited access to quality services and lack of screening facilities are major barriers in adequate detection and treatment of the disease. Funding gap for TB programme may increase these challenges. We have added the following lines in the discussion.

“It is evident that although the TB services are free in Bangladesh, access to quality services and modern screening facilities are major challenges in adequate detection and treatment of TB cases [31]. Funding gap may enhance such challenges, increase catastrophic expenditure [32], and decrease case notification as happened in 2011 due to the funding gap from GFATM [33]. However, we did not study the consequences of underfunding during 2019-2022. It would be interesting to assess the consequences of the funding gap in future studies.” Please see page 14, line 323-327

Other comments:

Line 67-72 -- Perhaps simply refer to Global Health Cost Consortium (https://ghcosting.org/), which summarizes the available cost data for tuberculosis. You can also look to Value TB studies. The current summation is inadequate.

Response: Many thanks for this suggestion. We have now added the reference. 

Lines 93-94 - unclear language "We collected facility-related data from purposively selected four types of twelve facilities representing different level of diagnosis and treatment of government and NGOs"

Response: We have now revised the lines as follows “We collected facility-related data from twelve purposively selected facilities representing different levels of diagnosis and treatment offered by the government and NGOs service providers. The facilities selected were from four main types of providers: 1) DOTs Centres (government and NGO), 2) chest disease clinics (CDC), 3) chest disease hospitals (CDH) (government and NGO), and 4) national and regional TB reference laboratories”

Results and Discussion -- these should be separate.

Response: We followed the journal suggested heading for formatting.

---

## [Editor Report · Decision Letter 2]

19 May 2023

Costs of services and funding gap of the Bangladesh National Tuberculosis Control Programme 2016-2022: An ingredient based approach

PONE-D-22-08101R2

Dear Dr. Hasan,

We’re pleased to inform you that your manuscript has been judged scientifically suitable for publication and will be formally accepted for publication once it meets all outstanding technical requirements.

Kind regards,

Kevin Schwartzman

Academic Editor

PLOS ONE

Additional Editor Comments (optional):

Thank you for addressing the reviewers' comments.
---

## [Editor Report · Acceptance letter]

24 May 2023

PONE-D-22-08101R2 

Costs of services and funding gap of the Bangladesh National Tuberculosis Control Programme 2016-2022: An ingredient based approach 

Dear Dr. Hasan:

I'm pleased to inform you that your manuscript has been deemed suitable for publication in PLOS ONE. Congratulations! Your manuscript is now with our production department. 

Kind regards, 

on behalf of

Dr. Kevin Schwartzman 

Academic Editor

PLOS ONE